# Stability Analysis of Multi-Layer Highwall Mining: A Sustainable Approach for Thick-Seam Open-Pit Mines

**Ya Tian** [1], **Lixiao Tu** [1,2], **Xiang Lu** [1,*], **Wei Zhou** [2], **Izhar Mithal Jiskani** [3], **Fuming Liu** [4] and **Qingxiang Cai** [1]

1   School of Mines, China University of Mining and Technology, Xuzhou 221116, China
2   State Key Laboratory of Coal Resources and Safe Mining, China University of Mining and Technology, Xuzhou 221116, China
3   NUST Balochistan Campus, National University of Sciences & Technology, Quetta 87300, Pakistan
4   Xinjiang Tianchi Energy Sources Co., Ltd., Changji 831100, China
*   Correspondence: xianglu@cumt.edu.cn

**Abstract:** Open-pit mining is a common method for extracting coal, but considerable resources are often left unrecovered at the bottom of end-slopes, leading to a permanent waste of resources. This research presents a sustainable approach of multi-layer highwall mining at different levels to address the issue of abundant resources left unrecovered at the bottom of the end-slope in thick-seam open-pit mines. The interlayer between the upper and lower entries is simplified into a beam structure model, the bending moment distribution characteristics of the beam under a load of highwall miner are analyzed, and a method for calculating the thickness range of the interlayer is proposed. The web pillar width and interlayer thickness, obtained theoretically, are verified through a numerical simulation, and the results of mining a single layer are compared to those of mining multiple layers. The results show that the web pillar width and interlayer thickness derived from the numerical simulation are basically the same as those of the theoretical analysis. Compared with single layer mining, the vertical stress on the web pillar in the lowest panel is reduced by 14.83~18.25%, and the safety factor of the web pillar is increased to 0.27. The web pillars and interlayers at different elevations are stable during multi-layer highwall mining. These findings support the feasibility of multi-layer highwall mining for resource recovery, which is conducive to sustainable mining.

**Keywords:** highwall mining; multi-layer mining; thick coal seam; beam structure model; coal pillar; safety factor

## 1. Introduction

Xinjiang is a major coal-producing region in China [1,2], with 23 open-pit coal mines, accounting for about 71.8% of the production capacity. The thickness of the main coal seam in these open-pit mines is generally above 10 m (Figure 1). Some mines can reach 80 m (Figure 2a). Although the region's favorable geological characteristics have brought substantial economic benefits [3], the coal resources below the end-slope cannot be excavated owing to the peculiarities of surface mining [4–6]. This phenomenon is more prominent in Xinjiang, where thick-seam open-pit mines dominate. It leads to a permanent waste of resources, which impedes the progression of green mining.

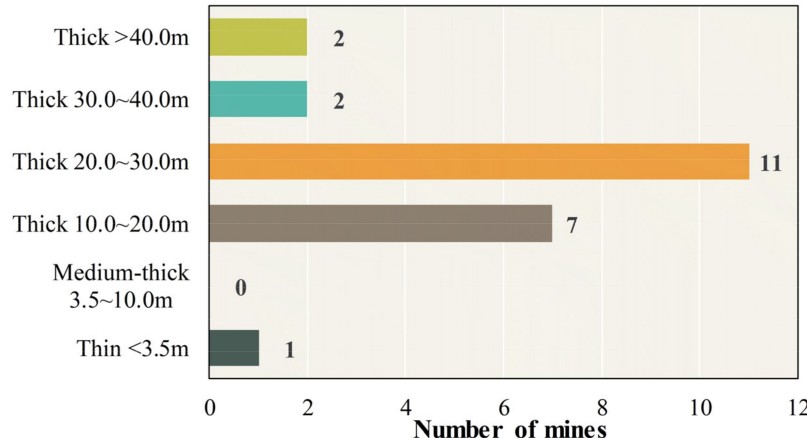

**Figure 1.** Thickness distribution of major coal seams in Xinjiang open-pit mine.

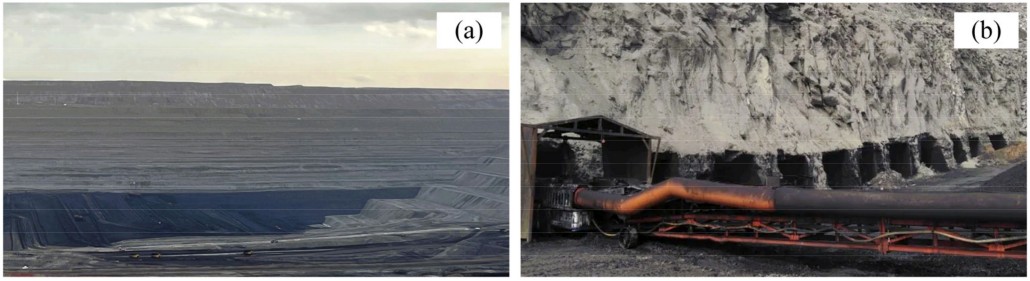

**Figure 2.** Description of the background. (**a**) Open-pit mine with coal seam thickness 80 m; (**b**) An early form of highwall mining in China [7].

Highwall mining can effectively exploit the resources [8] below the end-slope and extend the service life of the open-pit [9,10], which is essential for its sustainable development and for the practice of green mining. There are two types of methods for highwall mining. The first type is the auger system, which emerged in the 1940s in the United States of America (USA) and was initially used for contour mining outcrop coal [11]. This method is widely used because of its low price [12,13]. The second type is continuous highwall mining (CHM), which emerged in the USA in the mid-1970s and gradually replaced auger mining [14]. CHM is a remotely controlled drum shearer that cuts the end-slope coal seam to form a rectangular entry and transports coal resources out of the entry through a conveyor system [6,15]. CHM systems can produce 124,000 tons of coal per month [16,17], twice the capacity of auger production in the same geological conditions. This is the best option, except in weak geological conditions [14]. This method has been introduced globally and is implemented by many countries, such as Australia [18], India [19], Russia [20], and others [21,22]. In China, highwall mining was first described in the late 1990s, as pointed out in literature [23] that analyzes its potential application at the Antabao open-pit mine. In 2011, the Wulan Coal Mine in Inner Mongolia began to mine the end-slope coal resources with a road header and formed a simple highwall mining system with a belt conveyor (Figure 2b). The equipment has been continuously improved and popularized in open-pit mines in Inner Mongolia [24], Shanxi [25,26], and other places, in recent years.

Highwall mining has a long history of application and a broad application prospect in open-pit mines. Researchers in various countries have conducted corresponding studies on this approach, such as analyzing the mechanical strength of coal pillars following highwall mining to determine the appropriate width of coal pillars [27,28]. In addition, the literature elaborated on the influence of parameters such as the type of backfill material, the strength, and the filling rate on the stability of coal pillars inside the entry [29,30]. The algorithm proposed by Khayrutdinov et al. [31] for selecting the backfill material in terms of the rock stress-strain state is also of great importance for coal pillar stability control. In recent

years, the 3D similar physical simulation [32] and rock time effect [33,34] have also been explored in the technical background of highwall mining, and the research content has been further enriched. Ross et al. [35] focused on the highwall mining method for thick, steeply dipping coal seams in open-pit mines. It was suggested that multi-layer mining could be carried out at different elevations for sufficiently thick coal seams, with a vertical and thick interlayer between the entries above and below. However, when the mining height of the equipment is increased, the prior experience of retaining twice the mining height is regarded to be conservative, and numerical simulation is recommended to determine it. Wang [36] suggested that the interlayer thickness should be greater than the height of the mining roof arch with a certain safety distance, generally above 3.0 m. Newman and Zipf [37] used a simple beam model to propose an equation for the thickness and tensile strength of the interlayer, which is calculated over 0.45 m for the entry width of 3.16 m.

In the past, the study of multi-layer mining has been explored in various literature sources [38,39]. However, the current research delves into a crucial aspect that has been overlooked: the significance of vertical up-and-down web pillars and the adequate thickness of the interlayer during close-proximity multiple-seam highwall mining. If the pillars are not vertical and the interlayer is not thick enough (more than 4 m), the bottom of the interlayer will experience tensile damage as a result of the pressure from the upper part. Through the analysis of the maximum bending moment and tensile strength of the interlayer, the interlayer can be capable of bearing a load equivalent to roughly half of the tensile strength of coal. In addition, the prior research on multi-layer high-wall mining in the same coal seam has not considered the equipment load at the top of the interlayer, nor have numerical simulations been used to validate the theoretical calculations. This research aims to fill these gaps by utilizing a specific case of an open-pit end-slope mine with a coal seam thickness of 80 m. The beam structure model with applied equipment load is analyzed, and the range of the interlayer thickness is determined from the perspective of structural mechanics. The finite difference method software FLAC3D is used to calculate the models with different web pillar widths and interlayer thicknesses, which are used to verify the theoretical calculations. Finally, the stability of multi-layer highwall mining is discussed, providing a solid foundation for the design of highwall mining in thick coal seam open-pit mines. This, in turn, will contribute to sustainable mining by enabling the recovery of resources that are usually left unextracted at the bottom of end-slopes.

## 2. Engineering Background

### 2.1. Geological Conditions

This research is devoted to analyzing the stability of highwall mining. Therefore, the complex geological information of the open-pit mine under study was simplified. The open-pit mine is located west of the Zhundong coalfield in Xinjiang, where there is year-round drought, little precipitation, no surface runoff, and an undeveloped underground water system. The coal seam is 80 m thick and occurs nearly horizontally, while the overburden is siltstone with a thickness of 232 m. The end-slope has a total height of 312 m, a slope angle of 33°, and consists of 18 benches. The overburden benches are 15 m high and are divided into 30 m wide haulage benches and a 5 m wide catch bench. These two types of benches are arranged interlaced. The height of the coal benches is 30 m, and the width is 5 m (Figure 3).

### 2.2. Continuous Highwall Mining System

The highwall mining operations are performed using a remotely controlled EML 340 continuous mining machine developed by China Coal Science and Technology Cooperation (Figure 4), which forms a rectangular entry in the coal seam through a continuous miner or shearer. The rear of the shearer is overlapped with a multi-section structured belt conveyor to transport the coal resources out of the entry. The mining width is 3.3 m, the mining height is between 3.3 and 5.5 m, and the machine is capable of mining to a depth of more

than 300 m. The theoretical hourly production capacity is 15–27 t/h. The main parameters are shown in Table 1.

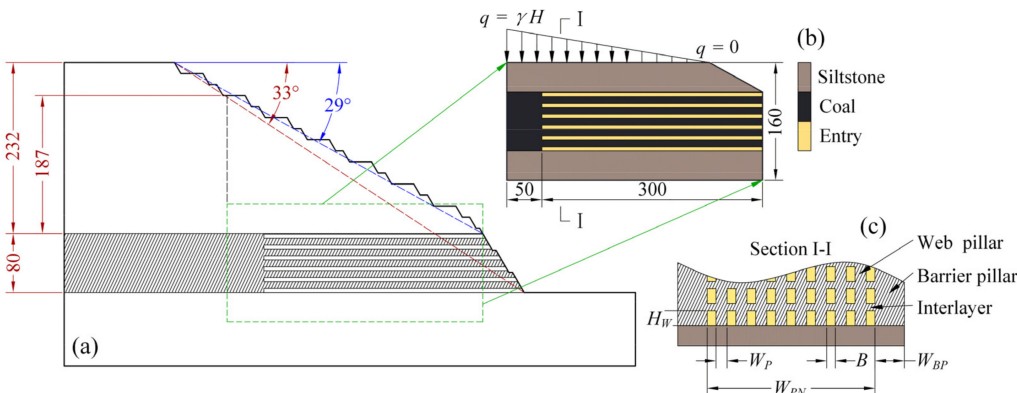

**Figure 3.** Open-pit slope and highwall mining characteristics. (**a**) End-slope morphology; (**b**) Multi-layer highwall mining simplified model; (**c**) Distribution forms and parameter symbols of coal pillars.

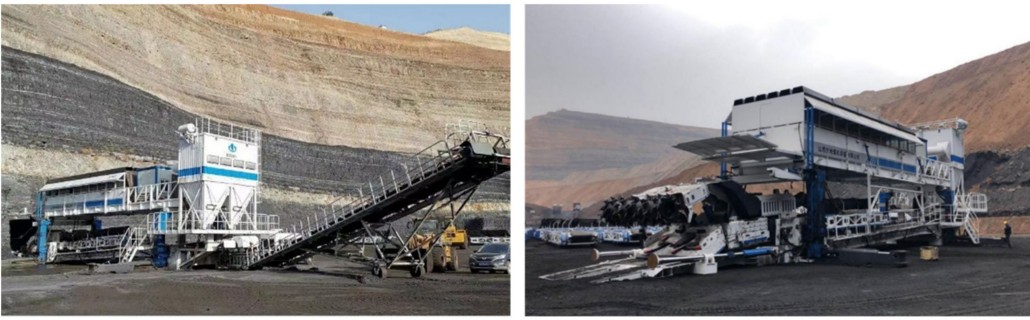

**Figure 4.** EML 340 highwall mining system [26,34].

**Table 1.** Parameters of highwall mining system.

| Parameters | Index |
|---|---|
| Dimension (length × width × height) (m) | $32.3 \times 11.7 \times 7.8$ |
| Weight (t) | 250 |
| Ground specific pressure (MPa) | 0.19 |
| Track width (mm) | 560 |
| Track center distance (mm) | 2019 |
| Mining width (m) | 3.3 |
| Mining height (m) | 3.3~5.5 |
| Adaptation of coal seam inclination (deg) | $+6\sim-16$ |
| Accuracy (m) | $\leq0.3$ (300 m) |
| Cuttable coal rock hardness (MPa) | $\leq40.0$ |

## 3. Highwall Mining Parameters

The parameters of highwall mining include the width of the web pillars and barrier pillars and the thickness of the interlayers. Among them, the parameters of the web pillars and barrier pillars are determined by the coal pillar strength and overburden load together. The strength of the long strip coal pillar during highwall mining is obtained using the Mark-Bieniawski empirical formula, and the overburden load of the coal pillar is determined by the tributary area method. The thickness of the interlayer is analyzed by constructing a mechanical model of the bending moment, and the range of the interlayer thickness is determined based on the bending moment and the tensile strength of the coal seam.

### 3.1. Coal Pillar Design

The web pillar strength is commonly obtained from the Mark-Bieniawski formula, which takes into account the effect of the web pillar length, which is more suitable for narrow and long rectangular web pillars [40].

$$S_P = S_I \left( 0.64 + 0.54 \frac{W}{H_W} - 0.18 \frac{W^2}{H_W L} \right) \tag{1}$$

where $S_P$ is the pillar strength, MPa; $S_I$ is the in situ coal strength, MPa; $W$ is the web pillar width, m; $H_W$ is the pillar height, m; $L$ is the pillar length, m.

As the length of a web pillar is generally much greater than its width and height, the equation can be simplified as:

$$S_P = S_I \left( 0.64 + 0.54 \frac{W}{H_W} \right) \tag{2}$$

The arrangement of web pillars during highwall mining is reasonably uniform and straight. Accordingly, the tributary area method considers that, after mining, the load originally borne by it will be transferred equally to the upper part of the web pillars on both sides. The weight of the direct roof caving directly above the entry is ignored in this approach, which simplifies the calculation. The self-weight stress field is the only source of the load on the upper part of the entry and web pillar.

$$S_{WP} = \frac{\gamma H (W + B)}{W} \tag{3}$$

where $S_{WP}$ is the vertical stress on the web pillar after highwall mining, MPa; $\gamma$ is the unit weight of the overburden, MN/m$^3$; $H$ is the thickness of the overburden, m; $B$ is the width of the entry, m.

The width of the entry is determined by the equipment, which is 3.3 m. Once the above parameters are determined, the safety factor of the web pillar ($SF_{WP}$) is obtained, which is generally required to be above 1.3 [14].

$$SF_{WP} = \frac{S_P}{S_{WP}} = \frac{S_I \cdot W (0.64 + 0.54 W / H_W)}{\gamma H (W + B)} \tag{4}$$

The design method of the barrier pillar is similar to the web pillar. Assuming that the number of web pillars in a panel is $N$ (generally in the range of 5 to 20), the width of the panel is:

$$W_{PN} = N(W + B) + B \tag{5}$$

The role of the barrier pillar is to divide the web pillars into various sections in order to prevent landslides or collapse caused by the interlocking instability of the web pillars in the panel. When all the barrier pillars in the panel are destroyed, the vertical stress on the barrier pillar is shown in (Equation (6)).

$$S_{BP} = \frac{\gamma H (W_{PN} + W_{BP})}{W_{BP}} \tag{6}$$

where $W_{BP}$ is the width of the barrier pillar, m.

The safety factor calculations for the barrier pillar are similar to the web pillar.

$$SF_{BP} = \frac{S_I \cdot W_{BP} (0.64 + 0.54 W_{BP} / H_W)}{\gamma H (W_{PN} + W_{BP})} \tag{7}$$

This method ignores the stresses of the web pillar in the panel, and the safety factor of the barrier pillar can reach a minimum value of 1.0, with a width-to-height ratio of the barrier pillar recommended to be greater than 3.0.

### 3.2. Interlayer Thickness Design

This section discusses the thickness of the interlayer between the mining entries using the structural analysis method. As shown in Figure 5a, when the continuous coal miner is mining from bottom to top, the forces on the interlayer are mainly the horizontal stress on both sides, the gravity of the interlayer itself, and the uniformly distributed load of the continuous coal miner. To simplify the calculations, the horizontal stresses and the self-weight of the interlayer are disregarded during the analysis. The web pillar, entry, interlayer, and continuous coal miner are reduced to a simple structural model (Figure 5b), which is considered a beam consolidated at both ends and subjected to two uniform loads with symmetrical sections. The track width ($W_a$) of the continuous coal miner is 560 mm, which is the range of the uniform load. The track center distance ($W_{cd}$) is 2019 mm, which is the distance from the center of the uniform load at both ends. The thickness of the interlayer is $H_B$, which is also the thickness of the beam in the structural model.

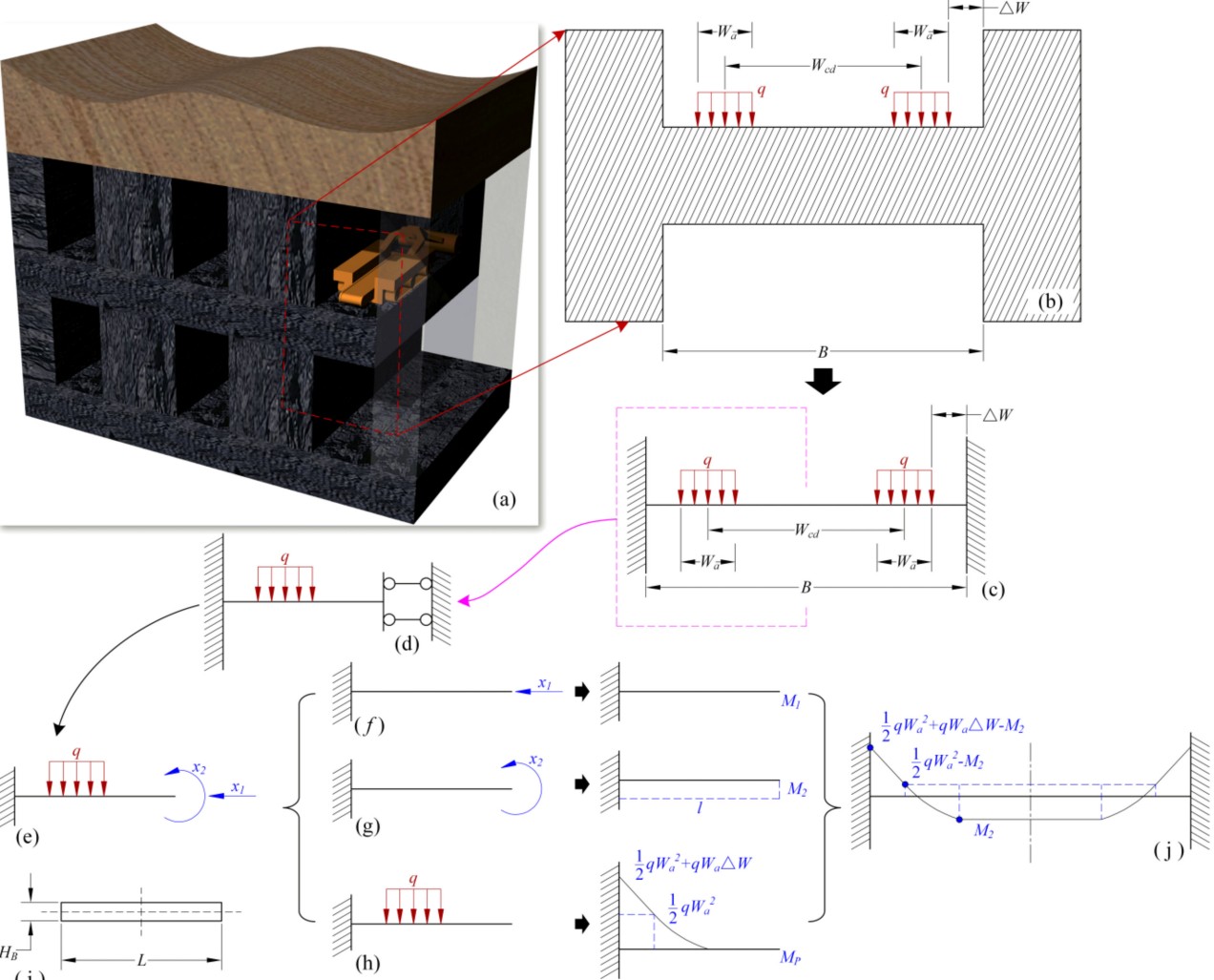

**Figure 5.** Design method for the thickness of the interlayer. (**a**) Diagram of multi-layer mining; (**b**) Force analysis of interlayer; (**c**) Force of original structure; (**d**) Equivalent structural forces; (**e**) Force of the basic structure; (**f**) Bending moment diagram of the basic structure under X1 load; (**g**) Bending moment diagram of the basic structure under X2 load; (**h**) Moment diagram of the basic structure under uniform load; (**i**) Section of the interlayer; (**j**) Original structural bending moment.

From the simplified structural stress diagram (Figure 5c), it can be seen that the interlayer is a symmetrical structure under a symmetrical load. It can be further simplified as a general structure with one end fixed, the other end as a sliding support, and only

subjected to a section of uniform load according to the symmetry (Figure 5d). The original cubic hyperstatic structure is simplified into a quadratic hyperstatic structure (Figure 5e), which reduces the number of unknown redundant forces. According to the force method, the displacement constraint at the sliding support can be released, and the corresponding unknown force can be applied. The force state comprises two kinds of unknown force constraints and a section of uniformly distributed load.

The force and bending moment diagram under the action of a single unknown force or load can be generated (Figure 5f–h) by analyzing the structural forces. In order to eliminate the disparity between the basic structure and the original structure, this section further establishes a set of displacement coordination condition equations.

$$
\begin{cases}
\delta_{11}X_1 + \delta_{12}X_2 + \Delta_{1p} = 0 \\
\delta_{21}X_1 + \delta_{22}X_2 + \Delta_{2p} = 0
\end{cases}
\tag{8}
$$

where $\delta_{11}$ is the displacement of the structure along the $X_1$ direction under the action of unit unknown force $X_1$, m; $\delta_{12}$ is the angle of rotation of the structure along the $X_2$ direction under the action of unit unknown force $X_1$, rad; $\delta_{21}$ is the angle of rotation of the structure along the $X_1$ direction under the action of unit unknown force $X_2$, rad; $\delta_{22}$ is the displacement of the structure along the $X_2$ direction under the action of unit unknown force $X_2$, m; $\Delta_{1p}$ is the displacement of the structure along the $X_1$ direction under the action of uniform load $q$, m; $\Delta_{2p}$ is the angle of rotation of the structure along the $X_2$ direction under the action of uniform load $q$, rad.

The calculation by the graphic multiplication method shows that:

$$
\begin{cases}
\delta_{22} = \dfrac{B}{EI} \\
\Delta_{2p} = -\dfrac{qW_a^3 + 3qW_a^2\Delta W + 3qW_a\Delta W^2}{6EI} \\
\delta_{11} = \delta_{12} = \delta_{21} = \Delta_{1p} = 0
\end{cases}
\tag{9}
$$

where $E$ is the modulus of elasticity of the interlayer, Pa. $I$ is the moment of inertia, m$^4$.

$$
I = \frac{LH_B^3}{12}
\tag{10}
$$

Substituting Equations (9) and (10) into Equation (8) shows that:

$$
\delta_{22}X_2 + \Delta_{2p} = 0
\tag{11}
$$

We obtain

$$
X_2 = \frac{qW_a(W_a^2 + 3W_a\Delta W^2 + 3\Delta W^2)}{3L}
\tag{12}
$$

Based on the superposition principle, the bending moment diagram of the structure under the joint action of the redundant unknown force and uniformly distributed load can be obtained (Figure 5j). To ensure the safety of the equipment during mining, the tensile stress on the interlayer under the action of the miner should be less than its tensile strength.

$$
\frac{M_{\max} \cdot c}{I} \leq \sigma_t
\tag{13}
$$

$$
M_{\max} = \frac{1}{2}qW_a^2 + qW_a\Delta W - M_2
\tag{14}
$$

where $M_{\max}$ is the maximum bending moment on the interlayer, kN*m; $M_2$ is the structural bending moment generated under the action of $X_2$, the same value as $X_2$, kN*m; $\sigma_t$ is the tensile strength of coal, taken as 0.4 MPa; $c$ is $1/2$ of the thickness of the interlayer $H_B$, m.



After sorting out these factors, Equation (15) can be obtained when the pillar length $L$ is taken as 300 m, and the calculated thickness of the interlayer is more than 1.73 m.

$$H_B \geq \sqrt{\frac{6M_{\max}}{\sigma_t L}} \qquad (15)$$

## 4. Numerical Simulations

A finite difference software, FLAC, which is a widely used geotechnical analysis, testing, and design tool [41], was used to carry out the numerical simulation. FLAC (Fast Lagrangian Analysis of Continua) is a numerical modeling code used for simulating the behavior of geotechnical systems, such as soil and rock masses, and the structures built on or within them. It uses the finite difference method to solve the equations of motion for the system, allowing for the analysis of complex behavior, such as nonlinear stress-strain relationships, anisotropy, and time-dependent behavior. FLAC can be used to simulate a wide range of geotechnical problems, including slope stability, underground mining, and tunneling.

The numerical models are divided into four categories (Figure 6): the web pillar model, the single layer model, the interlayer model, and the multi-layer highwall mining model. According to the relationship of the numerical analysis, the multi-layer highwall mining model is an integral model, while the remaining three numerical models are local. The local models verify whether the values of the highwall mining parameters are reasonable in a small area. The integral model is used to analyze the stability of the model when multiple parameters (web pillar width, interlayer thickness, number of mined seams) are applied to the same model. The web pillar model is established with 33,201 grid points and 30,000 zones. The length of the model is 350 m, and the length of the entry is 300 m. The $H_W$ is 5.5 m, $W$ is calculated according to Equation (4), and the $SF_{WP}$ is 0.5 to 2.0 (Table 2). The thickness of the overburden $H$ is taken to be 312 m. The web pillar is flanked by 1/2 of the entry, and 3 m of coal is retained above and below the pillar as a roof and a floor, respectively. The role of the web pillar model is to determine the safe range of values for $W$ by combining the theoretical calculation results.

**Table 2.** Calculation of web pillar width.

| $SF_{WP}$ | 0.5 | 0.75 | 1.0 | 1.15 | 1.3 | 1.45 | 1.6 |
|---|---|---|---|---|---|---|---|
| $S_P$ (MPa) | 6.85 | 7.61 | 8.54 | 9.05 | 9.64 | 10.23 | 10.74 |
| $S_{WP}$ (MPa) | 13.49 | 10.21 | 8.44 | 7.87 | 7.37 | 7.00 | 6.75 |
| $W$ (m) | 1.6 | 2.5 | 3.6 | 4.2 | 4.9 | 5.6 | 6.2 |

The single layer model consists of 480,074 grid points and 459,375 zones, with a height of 160 m, a length of 350 m, and a mining depth of 300 m. According to the calculation of the web pillar model in the later section, the $W$ is taken as 4.9 m, and the number of web pillars ($N$) in a panel is 9. The safety factor ($SF_{BP}$) is 1.0, while $W_{BP}$ is 19.0 m. On either side of the panel are 1/2 barrier pillars.

The interlayer model has 22,326 grid points and 18,000 zones. The central part of the model is a 3.3 m wide interlayer with thicknesses of 1.25, 1.50, 1.75, and 2.00 m. The top and bottom of the interlayer are empty, representing the mined entry. On both sides of the interlayer are web pillars of 5.5 m height and 4.9 m width. The upper part of the interlayer is subjected to a symmetrical uniform load of 190 kPa, and the depth of the model is 1.0 m in the mining direction.

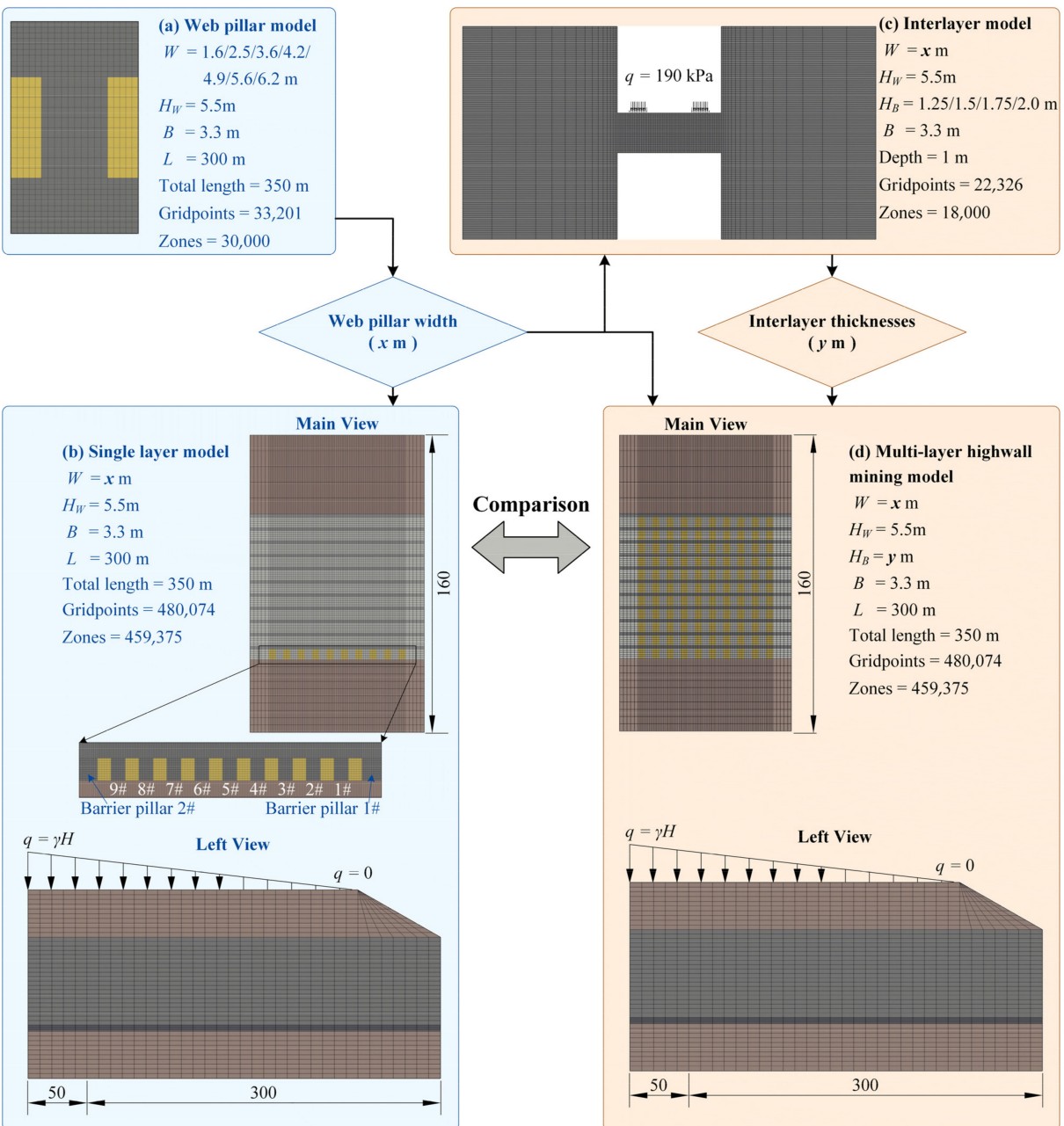

**Figure 6.** Flow chart of the numerical simulation modeling.

The multi-layer highwall mining model is obtained by simplifying the end-slope (Figure 3b), with 480,074 grid points and 459,375 zones. The model parameters are identical to the single layer model. Based on the local model calculation, the W is 4.9 m, the interlayer thickness ($H_B$) is 1.75 m, and the total number of mining layers is 11.

The model's parameters and the numerical simulation flow are shown in Figure 6. The model constrains the normal velocity around the perimeter and the velocity in all directions at the bottom before the calculation. Again, a linear load or a uniform load is applied at the top, depending on the depth at which the model is located. A linear elastic model is used for the siltstone on the top and bottom, while the strain softening Mohr-Coulomb model is used for the coal seam, the mechanical properties of which change with the strain. The physical and mechanical parameters of the materials are shown in Table 3. The mechanical parameters of the coal seam under different shear strains are shown in Table 4.



**Table 3.** Materials parameters used in the models.

| Rock Type | Cohesion $C_0$ (MPa) | Friction Angle $\varphi_0$ (deg) | Elasticity Modulus $E$ (GPa) | Poisson $v$ | Density $\rho$ (kg/m$^3$) |
|---|---|---|---|---|---|
| Siltstone | — | — | 0.672 | 0.262 | 2041 |
| Coal | 2.6 | 26 | 0.581 | 0.258 | 1248 |

**Table 4.** Change in $C_0$ and $\varphi_0$ with shear strain.

| Shear Strain | Cohesion $C_0$, MPa | Friction Angle $\varphi_0$, deg |
|---|---|---|
| 0.000 | 2.60 | 26 |
| 0.015 | 1.30 | 24 |
| 0.020 | 0.37 | 17 |

## 5. Results and Discussion

This section analyzes the web pillar, single layer, interlayer, and multi-layer highwall mining models and discusses the calculation results. The stability of the web pillars is analyzed by the variation law of vertical stress with mining depth. The strain increment in the vertical direction determines whether the interlayer thickness design is reasonable. The web pillar stability, stress state, and plastic zone distribution during multi-layer highwall mining are discussed to analyze whether the mining parameters determined by the modeling are applicable in the integral model.

### 5.1. Web Pillar

The simulation results of the web pillar model are shown in Figure 7. During the processing of the calculations, it was found that the deformation or displacement in the vertical direction is more uniformly distributed at the web pillar, with no discernible features. At the same time, considering that the stresses at the web pillar after highwall mining originate from the vertical direction, the stresses in the horizontal direction are lower and not very obvious in regularity. Consequently, the vertical stresses are analyzed. Figure 7a shows the vertical stresses at the central axis of the web pillars with different widths. The curves are divided into two types. When the $W$ is 3.6~6.2 m and the $SF_{WP}$ is 1.0~1.6, the vertical stress increases continuously with the mining depth. These curves are called "Type I" curves. When the $W$ is 1.6 and 2.5 m, the $SF_{WP}$ is 0.5 and 0.75, respectively. At this time, the stress that the web pillar can withstand is smaller. It fluctuates more with no obvious correlation to the mining depth. These curves are called "Type II" curves.

The web pillar profile at a mining depth of 276.32 m is further analyzed. Figure 7b shows the vertical stress at the center of the web pillar, and Figure 7c shows the vertical stress contours of the different $W$. The tributary area method can calculate the vertical stress that the web pillar should be stressed at different widths. The narrower the web pillar is, the greater the vertical stress is stressed, which develops as a power function. When the $W$ is 1.6 m, the vertical stress acting on it should be 13.49 MPa, which is much larger than the strength of the web pillar of 6.85 MPa at this time (Table 2). When the $SF_{WP}$ is less than 1, the web pillar is clearly fractured, and it cannot withstand higher stresses. The stress at the center of the web pillar is only 1.8 MPa, which is the residual strength of the damaged web pillar. When the width of the web pillar reaches 3.6 m, the $SF_{WP}$ is greater than 1.0, and the strength of the pillar is greater than the vertical stress applied.

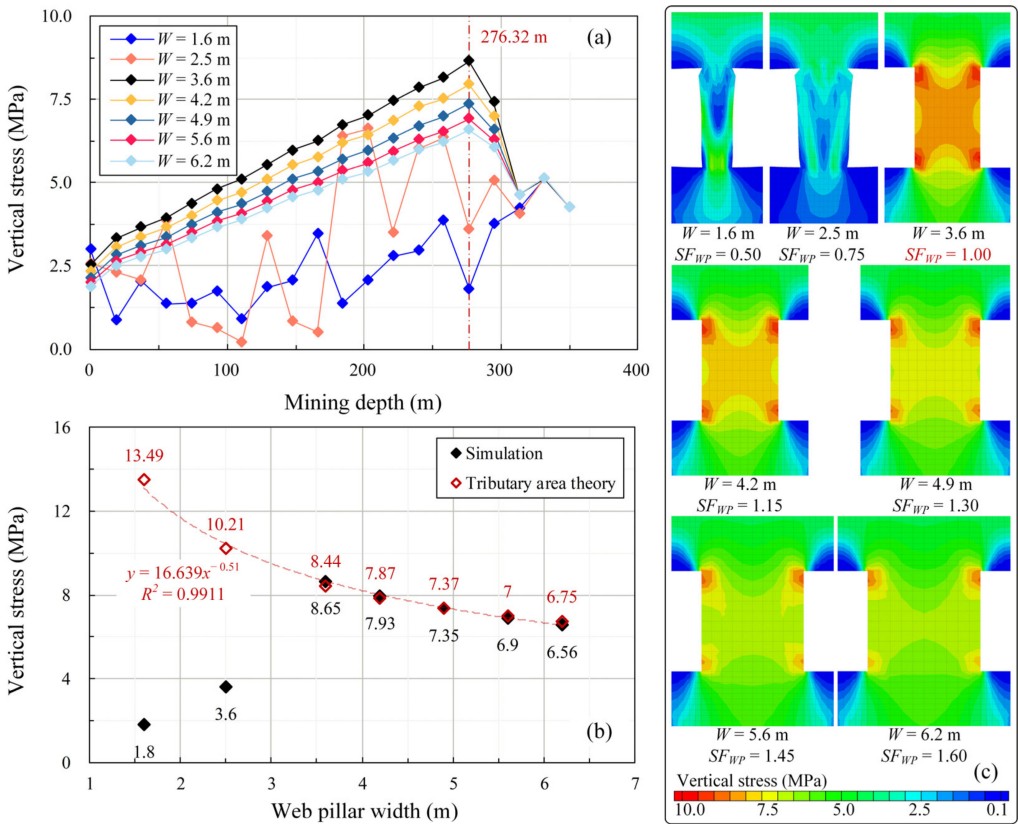

**Figure 7.** Simulation of web pillar model. (**a**) Vertical stress at the central axis of the web pillar; (**b**) Vertical stress at the center of the web pillar in a typical profile; (**c**) Vertical stress contour of the web pillar.

The vertical stresses on the web pillars obtained by simulation are almost identical to those obtained by the tributary area method. The ultimate strength of the web pillars is not reached, so they are in a stable state. When the web pillar is stable, the vertical stress distribution is "saddle-shaped", with large sides and a small middle. With the increase in the *W*, the stress distribution at the top of the web pillar gradually becomes nearly plateau-shaped (Figure 7c). Due to the presence of stress concentrations in the rectangular entry, the stress values are higher at the corners. When the web pillar is unstable, the vertical stress distribution is "V-shaped". The stress on both sides of the top of the web pillar is small, and the stress in the middle is large, which is obviously different from the stable state. These analyses demonstrate that the web pillar is stable when the vertical stress is distributed in a "Type I" curve. It is unstable if it is distributed in a "Type II" curve. The division of the two types of curves helps to evaluate the condition of the web pillar.

Figure 7c depicts the unstable damage mode of the 2.5 m wide web pillar. Figure 8 is further supplemented to illustrate the failure process of the web pillar. When the model begins to calculate, the mining excavation has not yet significantly affected the stress distribution in the web pillar, and the model is uniformly stressed overall. When the calculation reaches 25,000 steps, the stresses above and below the entry decrease significantly and gradually shift in the direction of the web pillar, and the concentration of the stresses around the entry becomes visible. As the computation approaches 60,000 steps, the vertical stress distribution inside the web pillar is "saddle-shaped". However, because the strength of the web pillar (7.61 MPa) is lower than the vertical stress at this time, the "saddle-shaped" distribution is only a transitional state. It exists only for a short time before it changes to the "8-shaped" distribution at 80,000 steps. The stress on the central point of the pillar is 8.38 MPa, and the stress on the top of the pillar is 10.17 MPa, so all of the load on the upper part of the original entry is carried by the pillar. From the stress

curve, "8-shaped" is the critical state when the "saddle-shaped" stable state changes to the "V-shaped" damage state. After the calculated number of steps exceeds 80,000, the web pillar is rapidly damaged, and the vertical stresses applied are reduced. When the calculation reaches close to 110,000 steps, the model unit is damaged due to excessive deformation, which leads to the termination of the calculation. The stress state of the web pillar could not be recorded after that.

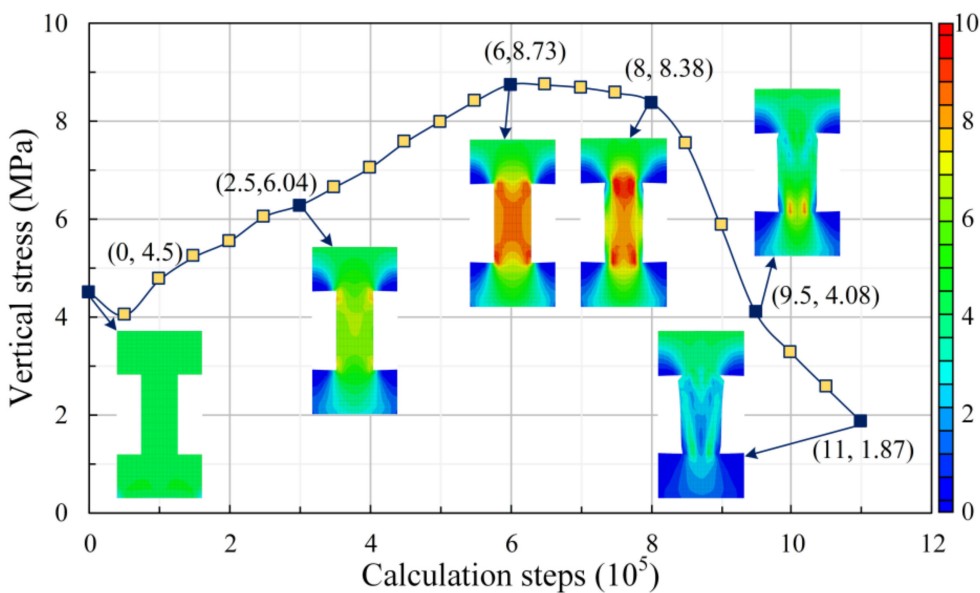

**Figure 8.** Damaging process of the web pillar (*W* = 2.5 m).

## 5.2. Single Layer

An analysis of the single layer mining model is performed to determine whether a 4.9 m wide web pillar can be used for the highwall mining of a single layer. The calculation results are shown in Figure 9. The model has a symmetrical structure, and the stress variation of the adjacent web pillars is small. Figure 9a only shows the relationship between the vertical stress (center position) and mining depth for 1, 3, and 5# web pillar and 1# barrier pillar. All four pillars are subjected to vertical stresses with a "Type I" curve distribution, and the web pillars are stable. Figure 9b shows the maximum vertical stresses and contours for each web pillar. From the vertical stress contour, we can also conclude that all the web pillars in the panel are stable, the web pillars from 3 to 7# are "saddle-shaped" in distribution, and the web pillars on both sides are gradually transitioning to "saddle-shaped".

The web pillar in the panel is stable, and the vertical stress on the barrier pillar is only 1.14 times before mining. The vertical stress of the web pillars on both sides of the panel is small and gradually increases towards the middle. The vertical stress on the web pillar in the panel is increased by 2.7~9.3% compared with the single web pillar when mining in a single layer. There is a small deviation from the results obtained from the web pillar model. The $SF_{WP}$ is reduced to 1.20~1.27. Therefore, it is reasonable to design with a 1.30 safety factor.

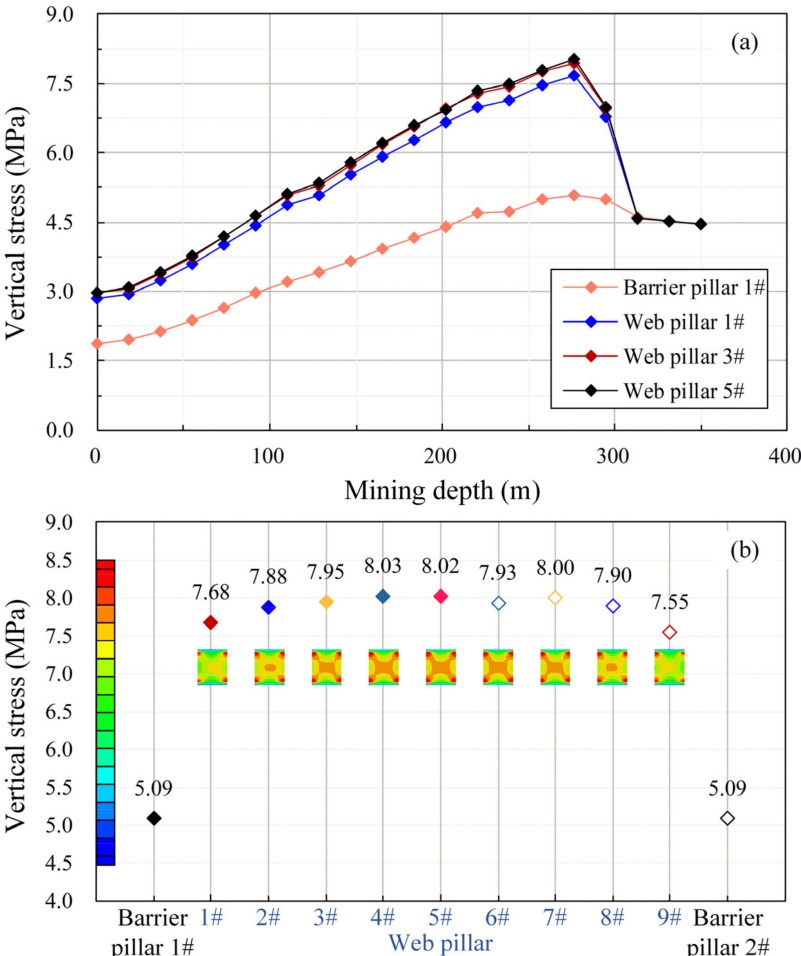

**Figure 9.** Single layer web pillar stress distribution ($W_P$ = 4.9 m). (**a**) Vertical stress at the central axis of the 1, 3, and 5# web pillar and 1# barrier pillar; (**b**) Maximum vertical stresses and contours for web pillar.

### 5.3. Interlayers of Varying Thickness

The vertical and horizontal stresses in the interlayer are relatively smaller than those of the web pillars on both sides, so it is difficult to carry out a comprehensive analysis. This section focuses on the strain increments of the unit in the vertical direction to describe the tensile or compressive conditions of the entire interlayer with web pillars on both sides as the continuous highwall miner passes by. Finally, the analytical findings of the vertical strain increments are demonstrated through the plastic zone distribution of the model.

Figure 10a–d shows the strain increment distribution of the model when the $H_B$ is 1.25, 1.5, 1.75, and 2.0 m. The part of the strain increment greater than 0 is red, indicating that the unit is under tension. The portion with a strain increment less than 0 is green, indicating that the unit is under compression. The higher the absolute value of the increment, the more obvious the tensile (compressive) stress concentration. The larger the thickness of the interlayer, the smaller the strain increment in the vertical direction. When the $H_B$ is 1.25 m, there is a concentration of tensile stress in the middle of its lower side. The strain increment in the vertical direction is close to 0.05, while the deformation in the other zones is relatively small. When the $H_B$ is 1.50 m, the tensile stress concentration in the center of the lower part of the interlayer still exists, but the magnitude is weakened. It is possible to observe the pressure on the lower compartment of the miner, the pressure on the top corner point of the entry, and the tension in the bottom corner point. With the further increase in the $H_B$, there is no tensile stress concentration at the lower part of the interlayer. The stress state of the model is clearly defined, and the compressive stresses at the corner point on the

upper side of the entry still interact with the compressive stress exerted by the miner load. This effect is further reduced when the $H_B$ is 2.0 m. The compartment strain increment is always $-2.5 \times 10^{-4}$ due to the miner load.

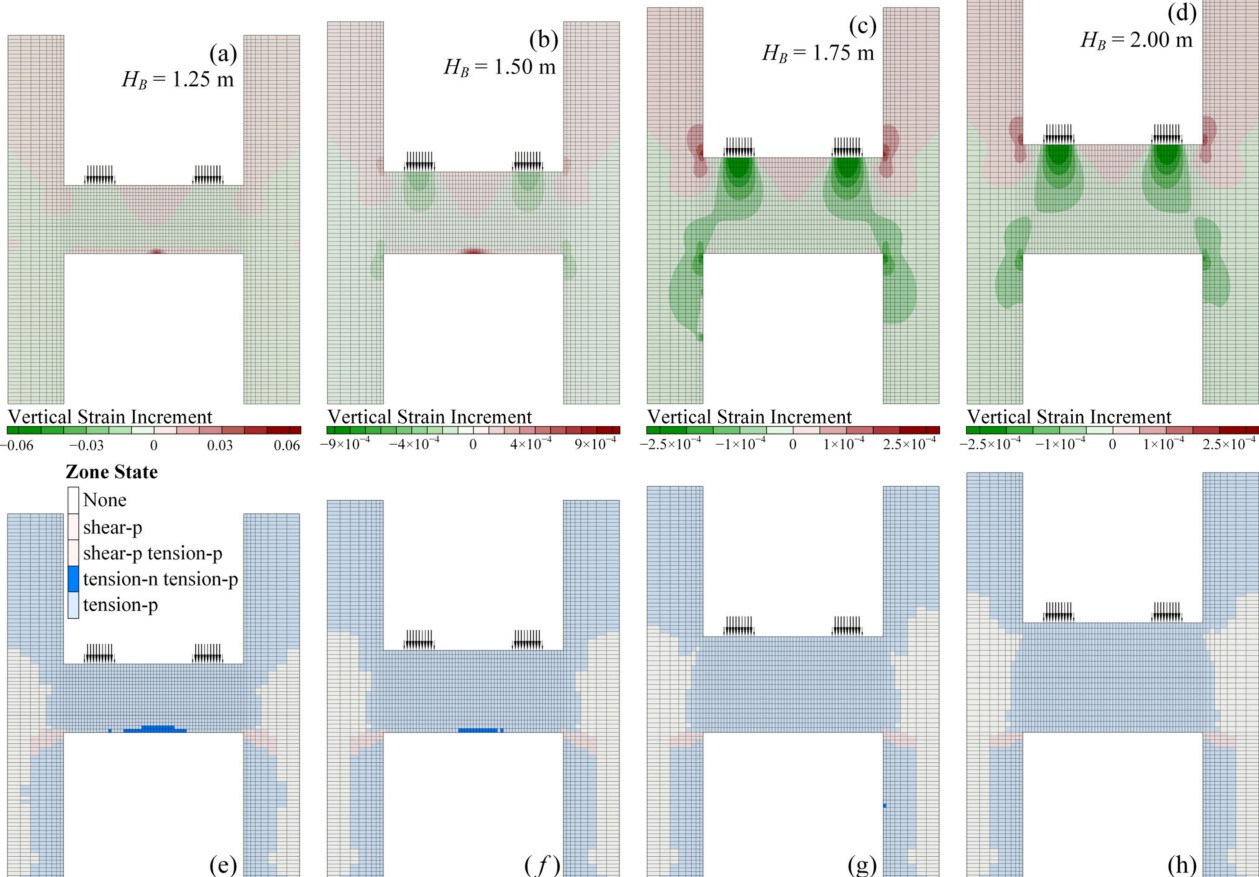

**Figure 10.** Distribution of vertical strain increment and plastic zone. (**a**) Strain increment distribution for different thickness of interlayer; (**b**) Plastic zone distribution for different thickness of interlayer. (**a**–**d**) shows the strain increment distribution of the model when the $H_B$ is 1.25, 1.5, 1.75, and 2.0 m. (**e**–**h**) shows the plastic zone distribution of the model with different $H_B$.

Figure 10e–h shows the plastic zone distribution of the model with different $H_B$. The plastic zone yields the unit by stress when the plastic constitutive model is used in FLAC. The software marks the unit as either a shear failure or tensile failure depending on the stress state when yielding occurs. The shear-p and tension-p in the legend indicate that the units are in a plastic state during the calculation. Due to the computational characteristics of the dynamic relaxation of the software, this part of the region has returned to an elastic state when the model is solved. Generally, only the shear-n and tension-n in the plastic flow state are analyzed. When the $H_B$ is 1.25 and 1.5 m, the unit at the middle side of the bottom of the interlayer is subjected to tensile damage, the $H_B$ increases, and the width of the tensile failure area decreases. The model is complete with no significant areas of plastic failure occurring for a $H_B$ of 1.75 and 2.0 m. From the distribution of the tensile failure plastic zone (tension-n), it can be confirmed that the smaller the thickness of the interlayer, the more obvious the phenomenon of the tensile stress concentration at the bottom. When the $H_B$ is 1.5~1.75 m, the tensile stress concentration below the interlayer is weakened, and no more damage occurs by tensile action. The results derived by Equation (15) have some reliability.

Further analysis of the vertical strain increments at the bottom of the interlayers with thicknesses shows that when the $H_B$ is less than 1.50 m (Figure 11a–d), the strain increment is generally positive, and the bottom of the interlayer is under tension as a whole.

Furthermore, the tension in the middle is more obvious, and the strain increment is larger. When the $H_B$ is 1.75 and 2.0 m (Figure 11e,f), the vertical strain increments at the bottom are negative, and the units are mainly under compression. The deformation produced by the pressure on both sides near the web pillar is relatively large. Then, it decreases rapidly on both sides of the interlayer. As the monitoring point moves from the left to the right of the bottom of the interlayer, the strain increment shows an increasing and then decreasing variation law, forming a symmetrical arch. The middle of the interlayer changes from tensile to compressive.

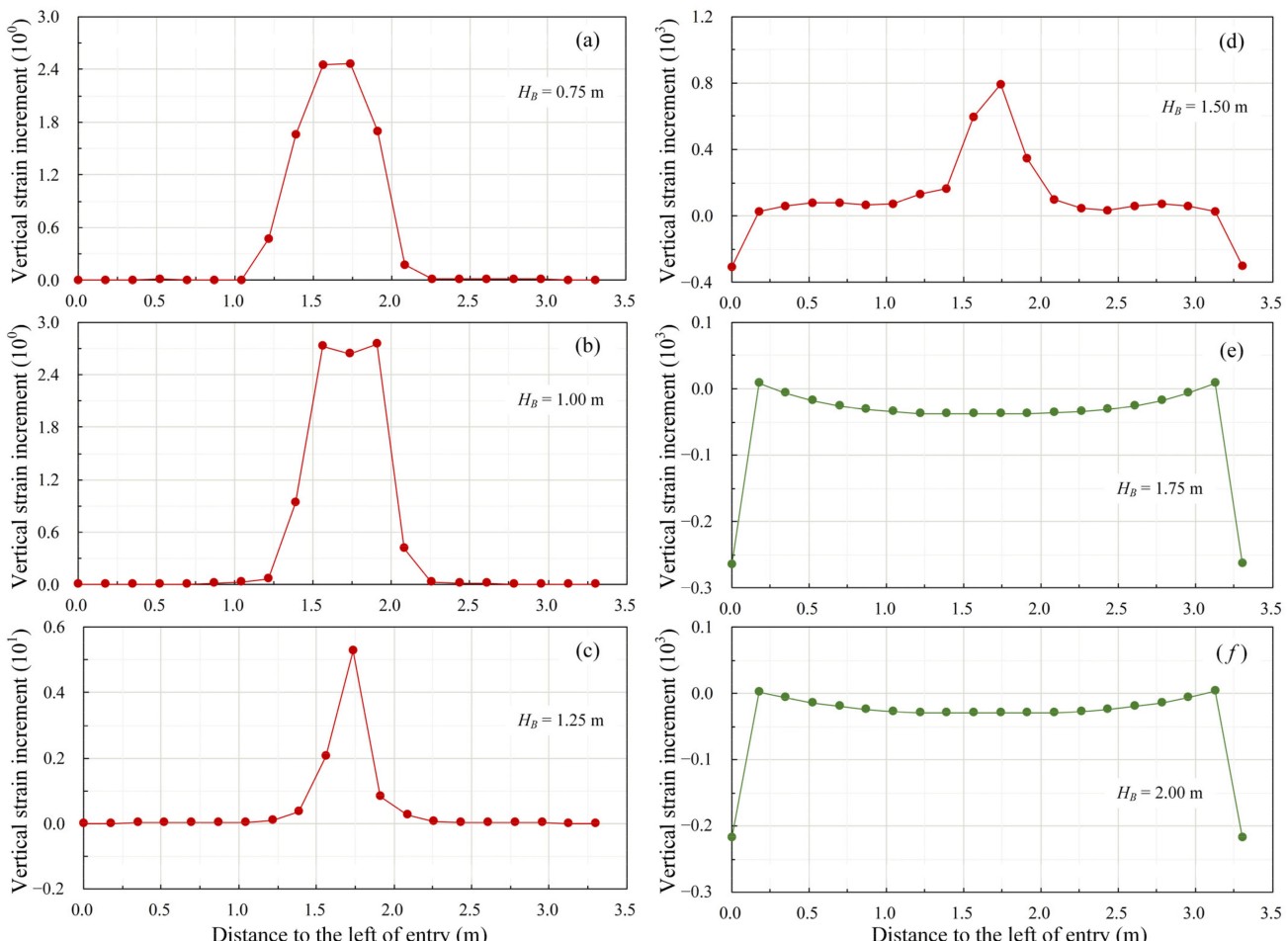

**Figure 11.** Vertical strain increment of different interlayer thicknesses. (**a**) $H_B = 0.75$ m; (**b**) $H_B = 1.00$ m; (**c**) $H_B = 1.25$ m; (**d**) $H_B = 1.50$ m; (**e**) $H_B = 1.75$ m; (**f**) $H_B = 2.00$ m.

### 5.4. Highwall Mining of Multi-Layer

The simulation results of the multi-layer highwall mining model are shown in Figure 12. Figure 12a shows the vertical stress distribution at the mining depth of 276 m. When the mining height is 5.5 m, and the interlayer height is 1.75 m, a total of 11 mining layers can be mined for an 80 m thick coal seam. This section names the number of mining layers m from bottom to top, where the lowest layer m = 1 and the top layer m = 11. As shown in Figure 12a, the vertical stress of the web pillar increases from top to bottom, layer by layer. The maximum vertical stress of the whole profile is at the corner point of the 5# web pillar in the first layer. The minimum vertical stresses are distributed between the upper and lower entries, with 0.03 to 1.0 MPa of vertical stress in the interlayer.

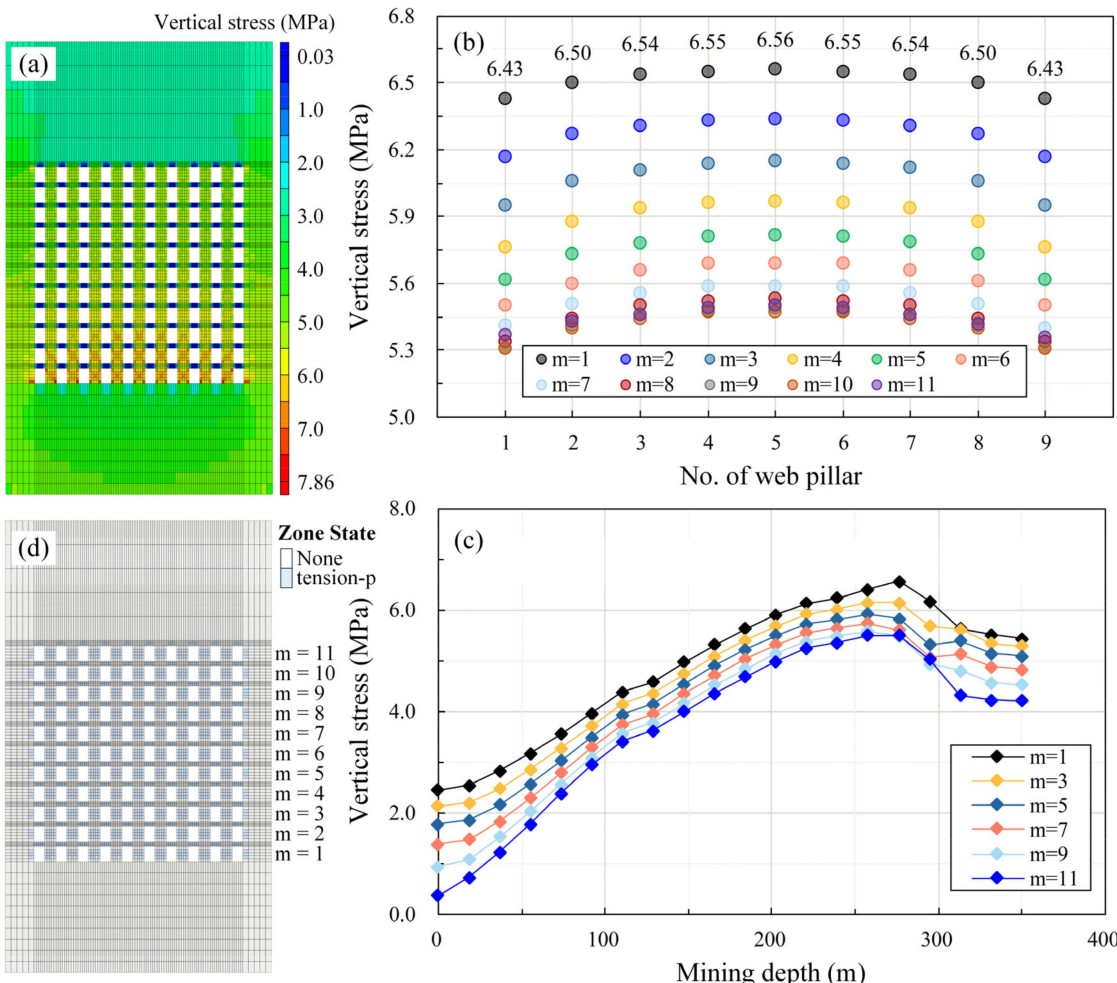

**Figure 12.** Results of the multi-layer highwall mining model. (**a**) Vertical stress distribution at mining depth of 276 m; (**b**) Vertical stress at the center of web pillars at different levels; (**c**) Variation of vertical stress with the mining depth for 5# coal pillars at different levels; (**d**) Plastic zone distribution at mining depth of 276 m.

Figure 12b shows the vertical stresses at the center of the web pillars at different levels. The closer the web pillar of the same layer to the middle, the greater the vertical stress it is subjected to, which is the same as the stress distribution law when mining a single layer (Figure 9). However, the vertical stress on the lowest web pillar when mining multi-layers is reduced compared with one layer (Table 5), with a reduction from 14.83 to 18.25%, and the stability of the lowest 5# web pillar is improved from 1.20 to 1.47. It is thus clear that multi-layer mining can reduce the vertical stress on the lower web pillar and improve the stability of the web pillar. When the highwall mining is from top-to-bottom, the vertical stress that the lowest web pillar can be subjected to decreases. It provides the conditions for further reducing the *W* and increasing the recovery rate, which is an essential indicator of green mining. When the web pillar layer is more downward (m = 1~6), the difference in the vertical stress between adjacent layers is larger. When *m* is in 8~11, the vertical stresses on the web pillars are similar, and the difference in the vertical stresses between layers is small.

**Table 5.** Vertical stress of web pillar under different mining conditions (m = 1).

| Web Pillar | 1# | 2# | 3# | 4# | 5# | 6# | 7# | 8# | 9# |
|---|---|---|---|---|---|---|---|---|---|
| Single layer (MPa) | 6.43 | 6.5 | 6.54 | 6.55 | 6.56 | 6.55 | 6.54 | 6.5 | 6.43 |
| Multi-layers (MPa) | 7.68 | 7.88 | 7.95 | 8.03 | 8.02 | 7.93 | 8.00 | 7.9 | 7.55 |
| Reduced value (MPa) | 1.25 | 1.38 | 1.41 | 1.48 | 1.46 | 1.38 | 1.46 | 1.4 | 1.12 |
| Reduction rate (%) | 16.27 | 17.51 | 17.73 | 18.43 | 18.20 | 17.40 | 18.25 | 17.72 | 14.83 |

For the convenience of observation, Figure 12c shows the variation curve of the vertical stress with the mining depth for 5# web pillar in some layers. All six curves are "Type I" curves, and the web pillar is stable. Figure 12d shows the distribution of the plastic zone of the model at the mining depth of 276 m, in which the model is complete, the unit state is only "None" and "tension-p", and the phenomenon of unit yielding does not occur. The mining parameters obtained from the local model can be applied to the integrated model. The model is complete in multi-layer mining, with no damage to the web pillars and the interlayer. Therefore, the model of multi-layer highwall mining can be used to analyze similar studies.

## 6. Conclusions

The current research is based on the premise that the end-slope should be mined in a bottom-to-top sequence while considering the operational parameters of the highwall mining equipment. A beam structural model with both ends fixed is developed based on the operational parameters, which include the continuous coal miner load, entry width, and interlayer thickness. The distribution of the force and bending moment in the structure was obtained using the force method and graphic multiplication method. The relationship between the equipment load and the interlayer thickness is derived, and a method for determining the interlayer thickness is proposed. Numerical simulations are performed to analyze the stress state of the web pillar with varying widths. The conclusions drawn from the theoretical calculations and numerical simulations indicate that when the width of the web pillar exceeds 3.6 m, the safety factor increases to 1.0. The vertical stress shows a "Type I" curve that first increases, and then decreases, with the increasing mining depth. The vertical stress at the peak is distributed in a "saddle shape" when the web pillar is stable. Conversely, if the safety factor falls below 1.0, the web pillar becomes unstable, leading to a "Type II" curve with irregular variations in the vertical stress and "V-shaped" stress distribution in the profile. The critical thickness of the interlayer damage was between 1.5 and 1.75, consistent with the theoretically derived result of 1.73 m. At an interlayer thickness greater than or equal to 1.75 m, the tensile stress concentration on the bottom side dissipates, and the stress state changes from tensile to compressive. In the multi-layer highwall mining model with a web pillar width of 4.9 m and an interlayer thickness of 1.75 m, stability was achieved for both the web pillar and the interlayer. The vertical stress on the web pillar was higher in the middle of the same panel and gradually decreased towards the sides. The vertical stress on the lower web pillar was reduced by 14.83~18.25% compared to single layer mining, resulting in an improved safety factor of 0.27. These results provide valuable insights into the design and optimization of high-wall mining operations. The multi-layer highwall mining model presented in this study has demonstrated that it can effectively reduce the vertical stress on the web pillar and improves the stability, thereby facilitating the sustainable development of mines, provided that the design of the mining parameters is reasonable with mutually compatible technology.

**Author Contributions:** Conceptualization, W.Z. and F.L.; methodology, Y.T. and X.L.; software, Y.T.; validation, Y.T. and I.M.J.; formal analysis, L.T.; investigation, Y.T. and W.Z.; resources, X.L.; data curation, X.L.; writing—original draft preparation, L.T.; writing—review and editing, Y.T. and I.M.J.; visualization, Y.T. and L.T.; supervision, Q.C.; project administration, F.L.; funding acquisition, X.L. All authors have read and agreed to the published version of the manuscript.

**Funding:** This research was funded by the National Natural Science Foundation of China (52204159) and the Natural Science Foundation of Jiangsu Province, China (BK20221125).

**Institutional Review Board Statement:** Not applicable.

**Informed Consent Statement:** Not applicable.

**Data Availability Statement:** The data that support the findings of this study are available from the corresponding author upon reasonable request.

**Conflicts of Interest:** The authors declare that they have no conflict of interest.

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
