# Peer review of "Stability Analysis of Multi-Layer Highwall Mining: A Sustainable Approach for Thick-Seam Open-Pit Mines"

_sustainability, doi:10.3390/su15043603_

Round 1

Reviewer 1 Report

Please check the attachment

Reviewer 2 Report

-         The literature survey is the main concern. New related papers in the studied field should be discussed and then the contribution of the current study should be highlighted.

-         Figure 1 should be divided into two separate Figures. Graph (a) in one figure and photos (b) and (c) in another.

-         In the numeral modeling section, all modeling steps should be presented and discussed.

-         In the simulation of the web pillar, what about the vertical displacement and horizontal stress at the center of the web pillar?

-         The sensitivity analysis of mechanical properties of coal seam and geometrical specification of extraction area on the stress and strain behavior of the pillars should be explained.

- The conclusion section is too long. It should present the main findings of the paper.

-         The whole text of the paper should be checked and revised for grammatical errors. 

Reviewer 3 Report

In this paper four different numerical approaches are used to analyse the stability of web pillars (inner pillars) a) web pillar model, b) interlayer model c) single layer model and d) multilayer model. The 2 last methods were used to define the barrier pillars' stability. The paper is well-written and all the relative references are cited. They have used a common numerical methodology to analyze the highwall coal mining method which lacks novelty but is interesting from the point of view that they have examined different models to derive a safe conclusion for the stability of the method based on pillar dimensions. Based on these comments, I suggest accepting this paper after minor corrections.    

Reading your paper, the method you used is well described. I have only a few comments to make to improve your work:

·       Figure 2: Both a) and b) in the figure’s caption.

·   You can use nomenclature to define some basic terms used in your paper such as web pillar, barrier pillar, Type I, II curves etc.

·   In a paragraph of lines 280-288 the safety factor estimated by theoretical analysis or based on the numerical model solution?

·       In figure 6 for the case of SF<1 obviously the pillar is fractured so high stresses can’t be endured.

·    In the conclusion of the analysis which of the (4) models do you propose to be used for the analysis of similar cases?

Round 2

Reviewer 1 Report

The manuscript "Stability Analysis of Multi-Layer Highwall Mining: A Sustainable Approach for Thick-Seam Open Pit Mines" by Ya Tian, Lixiao Tu, Xiang Lu, Wei Zhou, Izhar Mithal Jiskani, Fuming Liu, Qingxiang Cai was submitted for second review.

As can be seen from the submitted manuscript and the explanatory note to the review, the authors did a lot of work to make changes in accordance with the comments.

The revised manuscript is a completed scientific study on a highly relevant topic: analysis of the stability of mine workings in pit reserves development. The revised version of the manuscript, in my opinion, fully satisfies the requirements of a scientific article and can be published in the open press.

Reviewer 2 Report

no comment